# EXPLICIT RECALL FOR EFFICIENT EXPLORATION

## ABSTRACT

In this paper, we advocate the use of *explicit memory* for efficient exploration in reinforcement learning. This memory records structured trajectories that have led to interesting states in the past, and can be used by the agent to *revisit* those states more effectively. In high-dimensional decision making problems, where deep reinforcement learning is considered crucial, our approach provides a simple, transparent and effective way that can be naturally combined with complex, deep learning models. We show how such explicit memory may be used to enhance existing exploration algorithms such as intrinsically motivated ones and count-based ones, and demonstrate our method's advantages in various simulated environments.

## 1 INTRODUCTION

Incorporating deep neural networks (DNNs) has brought significant advance to reinforcement learning (Mnih et al., 2015; 2016). In addition to the ability to approximate nonlinear functions, convolutional neural networks (CNNs) provide an effective way to model local patterns, thus enabling the agent to learn from high-dimension visual inputs (*e.g.*, Atari 2600 games (Mnih et al., 2015)) or even combinatorial discrete spaces (*e.g.*, the game of Go (Silver et al., 2017)). However, as the scale of problems increases, efficient *exploration* becomes even more challenging. For instance, in the game Montezuma's Revenge (Bellemare et al., 2013), the player moves in a large partially-observable maze, searches for props and fights with monsters. Nonzero rewards come from successfully obtaining props or killing monsters that involve a long sequence of actions. As a result, nonzero reward signals in these problems are very sparse, and simple exploration algorithms like $\epsilon$-greedy are likely to fail.

How to exploit the power of DNNs for efficient exploration has become an active research field in Deep Reinforcement Learning (DRL). Sharing the same spirit of encouraging novelty, the majority of works in this track can be roughly divided into two groups. The first group of works, partially motivated by cognitive sciences, introduces the concept of novelty about states (Houthooft et al., 2016; Bellemare et al., 2016; Pathak et al., 2017). The agent will get extra "intrinsic" rewards when it explores "novel" states following some metrics. This is also called "curiosity-driven" exploration in some literatures (Pathak et al., 2017). The second group of work aims at introducing "goal-driven" exploration. The agent cooperates with a goal generator (either human-programmed oracles (Baranes & Oudeyer, 2013) or generators jointly optimized with the agent (Held et al., 2017; Sukhbaatar et al., 2017)). Given such internal goals, the agent will get rewards when it manages to reach the goal. As the difficulty in reaching goals increases, the agent will learn more skills and explore more states that it could not reach before, leading to self-paced learning or curriculum learning (Held et al., 2017).

In this paper, we study another important factor in exploration — memory, which has received little attention in the past. In particular, we introduce *Explicit Recall*, a general mechanism for efficient exploration that can be naturally *combined* with many existing exploration algorithms, including several that have been mentioned above. The observed trajectories are explicitly memorized in a prioritized data structure. Different from typical DRL algorithms which memorize the value (novelty) of states implicitly in a black-box DNN and compose policies to explore the novel states, in Explicit Recall, the exploration trials are based on the recall of observed trajectories leading to previously visited states with high novelty. Inspired by cognitive science findings that memorization of observed trajectories helps human to reach previously visited states (Thirkettle et al., 2013), we study whether and how explicit recall improves the exploration in DRL.

We study the performance of Explicit Recall on two typical exploration tasks: knowledge/skill learning and online exploration/exploitation tradeoff. The first focuses more on the pure exploration and aims at learning knowledge (*e.g.*, the model of the environment) or skills (*e.g.*, the capability of reaching goal states) during the exploration process. The second focuses on balancing exploration and exploiting in a more typical reinforcement learning setting, with the goal of maximizing long-term rewards.

## 2 EXPLICIT RECALL

Explicit Recall integrates explicit memory of observed trajectories into deep reinforcement learning. The exploration of the agent begins with the recall of a trajectory leading to a previously visited state that is considered to be novel. This is based on the assumption that: *states next to novel states are usually novel states*, where the state locality is defined by the state transition function. Although the validation of this assumption depends on both the metric used for measuring novelty and the environment, we empirically demonstrate the performance of models that rely on this assumption in challenging simulated environments. With the recalled trajectory, the agent first follows the trajectory to reach the end state of the trajectory (a novel state). The real exploration of the environment then begins. The trajectory of this new exploration trial gets stored in the memory.

To make the exposition concrete, in most parts of the paper, we adapt the state novelty definition from curiosity-driven RL (Pathak et al., 2017). It should be noted that Explicit Recall may be used in combination with other exploration algorithms as well, as demonstrated later. As in previous work of Pathak et al. (2017), we have a dynamics function $\psi$ that is trained for the environment based on agents' experience. The agents also learns an inverse dynamics function $\phi$ — predicting the state transition in the feature space of $\psi$. The prediction error is treated as the novelty for states. One can easily choose other metrics such as psuedo-count (Bellemare et al., 2016).

From the view of exploration, this approach makes the agent focus more on pure exploration but less on how to reach novel states. For better intuition, imagine that you live in a village located at the center of a big map. The whole map can be divided into multiple regions. The central area of your village is what you are very familiar with. On the edge of the village are the places where you may have previously visited but not so often, and thus are unfamiliar. The places beyond the edge of the village are what you do not know about at all. Your exploration should begin with reaching the border of the village, and then explore the outer world. Analogous to this idea, the agent begins its exploration from a novel state (on the edge) by following a memorized trajectory.

The idea bears some similarity to the E[3] algorithm of Kearns & Singh (1998), although they focus on finite-state MDPs and can use an exact planner (such as value iteration) to reach novel states. Extending E[3] or the related Rmax algorithm (Brafman & Tennenholtz, 2001) to deep RL is tricky: instead of using tabular planning algorithms, one must use complex blackbox models, such as a DQN (Mnih et al., 2015) or a policy network (Mnih et al., 2016), to encode learned policies to reach novel states. In contrast, our method memorizes trajectories to reach novel states, which is a simple and transparent process.

### 2.1 EXPLICIT TRAJECTORY MEMORY

The trajectory is organized as a tuple consisting a sequence of states $\{s_i\}_{i=1}^{k}$ and the actions $\{a_i\}_{i=1}^{k=1}$ along this $k$-length trajectory $traj$, and a real-valued score $\texttt{score}(traj)$ for this trajectory. Since we are interested in the novelty of the state that this trajectory is leading to, we choose the novelty of the end state $\texttt{novelty}(s_k)$ as the score. If we are also considering environmental (extrinsic reward), a mixture of extrinsic rewards and the novelty is used:

$$\texttt{score}(traj) = \texttt{novelty}(s_k) + \lambda\,\texttt{stage\_avg\_reward}(traj) \qquad (1)$$

where $\lambda$ is a task-dependent hyper-parameter and $\texttt{stage\_avg\_reward}$ is a function similar to average extrinsic reward function over the trajectory, see details in Appendix B.

We construct the algorithm based on the following assumption: states next to novel states are usually novel states. Thus, we give priority to trajectories with high scores. We organize the trajectory memory as a prioritized pool *pool* with a fixed capacity $\texttt{capacity}(pool)$. As mentioned before, during the most recent exploration trial, new trajectories are recorded periodically and added to the

memory. When the size of the pool exceeds its capacity, the trajectory with smallest $\texttt{score}(traj)$ will be removed from the pool.

## 2.2 Exploration with Trajectory Recall

Exploration starts from choosing places of interest based on the memory. The trajectory with highest score is selected (and then removed) from the pool. The end point of the trajectory is the target (goal) state to be reached. We discuss the problem of reaching the target state based on the recalled trajectory in the next section. After reaching the state, the agent will begin its exploration. In this phase, one can incorporate any existing exploration algorithms.

A naive way for exploration is to perform random actions, which actually, shows acceptable results in our empirical study. We discuss several choices of the exploration method in Section 3.1.

## 2.3 State Reaching

We now consider the problem of reaching a target state introduced by a trajectory. For simplicity, we assume that the environment always starts from the same initial state, which is a typical setting in reinforcement learning. If the state transition is completely deterministic, we can directly perform the recalled action sequence of the trajectory $\{a_i\}$ and reach the designated target. However, this condition is not satisfied in many real-world applications due to the noise or partial observability of the environment.

To resolve the state reaching problem with recalled trajectory, we incorporate a PATH function. Given a *(starting state, target state)* pair, PATH function outputs a probability distribution over action space for the first action to take at state $s$ in order to reach state $s'$. Thus, we can generate a path from current state to the goal state by iteratively invoking the PATH function.

The pseudo-code of the full algorithm for state reaching is demonstrated in Algorithm 1 in Appendix C. In short, we iteratively pick states from the trajectory as a sub-goal. PATH function is invoked to reach this sub-goal. The step-distance in the trajectory between the current state and the sub-goal is much less than the length of the whole trajectory. When the agent reaches the sub-goal, we will randomly pick a next sub-goal for the agent. The advantage of this procedure is that it explicitly decomposes a long-term goal into short-term sub-goals.

To determine whether the sub-goal state has been reached or not, we measure the cosine distance between the current state and the sub-goal in feature space introduced by the inverse dynamics $\phi$. When the similarity exceeds a hyper-parameter $thresh$, we treat two states to be the same.

The PATH function is jointly optimized with the agent. We treat the learning of PATH function as another reinforcement learning problem. If the agent successfully reaches a designated sub-goal (*i.e.*, a path from the starting state to the target state has been found), the agent will be rewarded. The reward is determined by the cosine distance between the actual ending state and the sub-goal. Moreover, we give small penalty to the agent at each step to encourage it to reach the sub-goal as fast as possible.

## 2.4 Summary

To sum up, the Explicit Recall framework is composed of the following modules:

1. An inverse dynamics model $\phi$ as the feature encoder for the states.

2. An state prediction function works on the feature space of $\phi$ and is optimized by experiences. We define the prediction error of this function as the novelty of states.

3. A PATH function that is independently optimized. PATH function takes *(starting state, target state)* pair as input and outputs a probability distribution over action space for the first action to take. It is optimized by reinforcement learning algorithms (A3C (Mnih et al., 2016) in our case) during the state reaching phase of the exploration. When the agent achieves a sub-goal, it will receive a reward.

4. A prioritized pool for trajectories of fixed capacity. The priority is defined in Equation 1.

## 3 EXPERIMENTS

In the experiments section, we do thorough experiments in `Rooms` environments to show that our methods have advantages over baselines in exploration ability and efficiency, knowledge learning efficiency and task exploitation efficiency. By concatenating with different baselines, we show that our framework is flexible to integrate other exploration methods and achieve better results. We further extend our method to hard-exploring Atari games Montezuma's Revenge and PrivateEye, and achieves better results than the baselines. The implementation details of our method and hyper-parameters are included in Appendix E.

### 3.1 INCORPORATE BASELINES

In the exploration phase of our method, as stated in 2.2, one can incorporate any existing exploration algorithms. One choice is to use curiosity-driven (*curiosity*) method (Pathak et al., 2017), which can be easily applied to high-dimension inputs. In `Rooms` environments, we can easily counting the number of visits of each state using oracles, therefore count-based exploration bonus (*count*) method (Strehl & Littman, 2008) is another good choice.

Besides using these baselines as the explorer, by incorporating intrinsic motivated reinforcement learning algorithms (*curiosity, count*), we can apply their novelty estimator (*i.e.* ICM module (Pathak et al., 2017) and count-based bonus function (Strehl & Littman, 2008)) to the trajectory. Therefore we produce two variants of algorithms by incorporating *curiosity* (as *ours+cur*) and *count* (as *ours+count*). In the experiments below, we show that our methods consistently outperforms their baselines respectively.

### 3.2 ROOMS ENVIRONMENTS

`Rooms` are a set of environments extending the famous `FourRooms` environment (Sutton et al., 1999). To make the environments more challenging, we not only increase the number of rooms but also block the doors between the rooms to create hard-exploring environments which require a longer horizon to explore. We mainly use 4 variants of `Rooms` Environment with the same size $37 \times 37$ of the map (include borders), which are normal-shape $2 \times 2$ rooms (classical four-rooms environment), zigzag shape $2 \times 2$ rooms, normal shape $3 \times 3$ rooms and zigzag shape $3 \times 3$ rooms. The exploration becomes harder when the number of rooms increases or with the zigzag shape. We also show results on more challenging zigzag shape $6 \times 6$ rooms in Appendix D. The accurate definition of these variants and environment setting can be found in Appendix A.

As the locations are easy to be obtained from the environment oracle, facilitating computing the number of visits to states and monitoring the exploration progress, we do thorough experiments in `Rooms` environments and compare the exploration efficiency in different aspects.

#### 3.2.1 PURE EXPLORATION

To better study the exploration efficiency of our methods and baselines, we adopt the pure exploration setting where no external reward is provide. Empirically, we use a statistics similar to count-based exploration bonus (Thrun, 1992; Strehl & Littman, 2008) (which is equivalent to the sum of reward bonus for visiting all states once at that point):

$$S = \sum_{x \text{ is not wall}} \frac{1}{\sqrt{\text{visit\_times}[x] + 1}},$$

and the lower $S$ means the exploration is more sufficient where there are less less-visited states.

We show the results of *count*, *curiosity*, *ours+count* and *ours+cur* methods in four `Rooms` variants in Figure 1. We conclude that the exploration efficiency relation is roughly *ours+count* > *ours+cur* > *count* > *curiosity*. Our methods consistently outperform their baselines respectively, and stronger baselines bring a stronger performance of our methods.

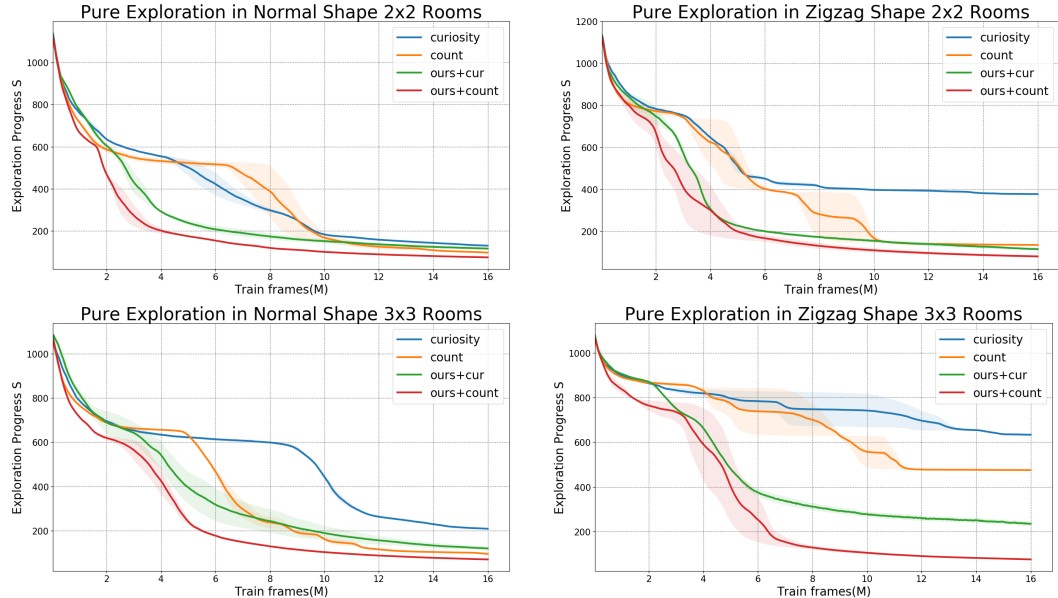

Figure 1: The exploration progress for *count*, *curiosity*, *ours+count* and *ours+cur* methods in four variants. Lower is better, indicates the distribution of number of visits is more smooth. By incorporating the *count* and *curiosity* baseline, our methods consistently outperforms the baselines, especially when the exploration becomes harder (zigzag shape $3 \times 3$ rooms). Generally, the *count* dependent ones (*ours+count*, *count*) outperforms its counterpart with curiosity, which is reasonable since they use oracle information.

### 3.2.2 KNOWLEDGE LEARNING EFFICIENCY

Not only able to learn the dynamics while exploring, more importantly, but our method is also able to learn task-independent skills (PATH function) might be useful for other tasks (such as following an unseen demo (Pathak et al., 2018)). Among different choices of the modeling, we choose inverse dynamics (Pathak et al., 2017) as the environment model, which has been proved to be an effective way of representing states under noisy environments. We directly extract the inverse dynamics model $\phi$ learned in *ours+cur* as well as *curiosity*.

Inverse dynamics is a function that given the current and next state $(s, s')$ pairs to output an action $a$ that was applied to the environment so that $env(s, a) = s'$. The inverse dynamics model $\phi$ is tested on all pairs of adjacent states $(s, s')$ that $s \neq s'$ to avoid ambiguity. We report the curve of accuracy in Figure 2.

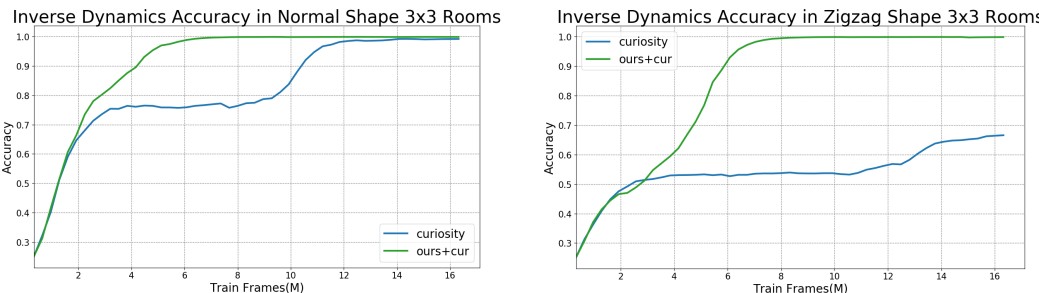

Figure 2: The accuracy curves of inverse dynamics models for *curiosity* and *ours+cur* method in normal (left) and zigzag (right) shape 3x3 rooms. Our method is more efficient in learning and learns better inverse dynamics models.

### 3.2.3 TASK EXPLOITATION EFFICIENCY

Intuitively, exploration efficiency and the task performance is correlated, especially when the reward signals are sparse. We extend the pure exploration to destination finding tasks by adding a fixed destination. Reaching the destination within limited steps gets a reward of value 100, otherwise no reward.

The results are shown in Figure 3. In these tasks, the performance is consistent with the pure exploration tasks, where the efficient-explore methods have higher chances being able to reach the destination.

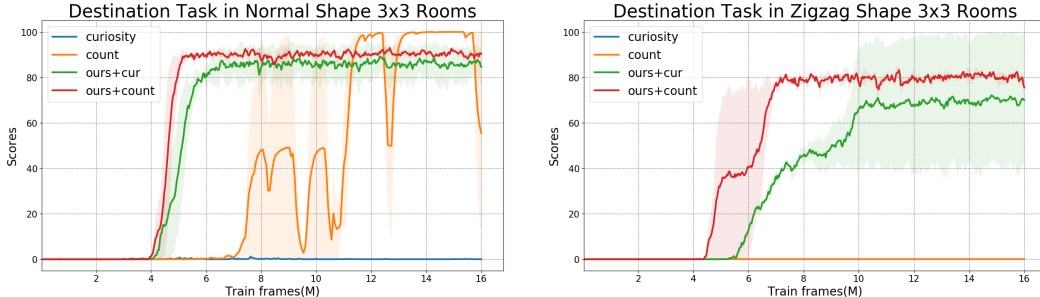

Figure 3: The training score of destination task for *curiosity*, *count*, *ours+count* and *ours+cur* methods in normal (left) and zigzag (right) shape 3x3 rooms. Our method is more efficient to reach the destination in both environments.

### 3.2.4 RESULT ANALYSIS AND VISUALIZATION

In Figure 4, we provide a series of snapshots from our video visualizing the exploration progress of our method. Our explicitly recalled target states (second row) are easy to interpret. Please visit our website [1] for more demo videos of the pure exploration tasks in Rooms environment variants.

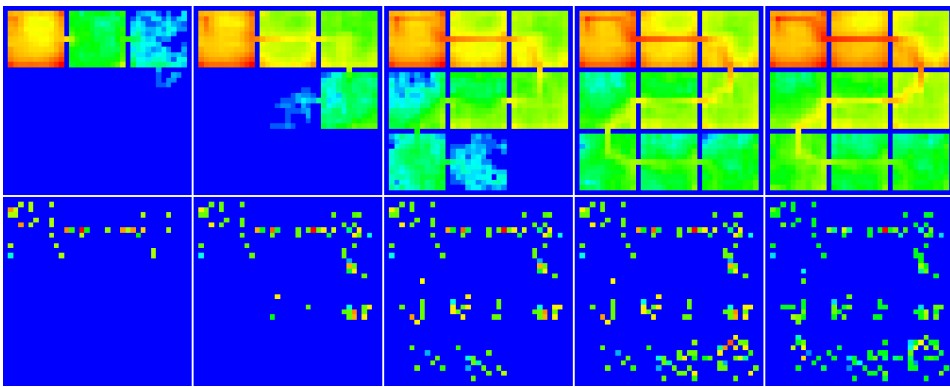

Figure 4: Visualization of the exploration progress in zigzag shape 3x3 rooms using *ours+cur* method. The $x$-axis time or the number of training frames. The top row is the heatmaps of the number of visits for states. The second row is the heatmaps for states that the number of times being selected as a target state. All heat maps are shown in log-scale. It can be seen that most of the chosen target states are located at the frontier of the known state space. Also, there exists a clear pattern in the first row showing routes used to reach the frontier.

### 3.3 ATARI GAMES

We extend *ours-cur* method to Atari games whose states are in high-dimensional space. We choose two games which are known to be hard-exploring (Bellemare et al., 2016), Montezuma's Revenge and PrivateEye, to test the exploring ability of our methods.

---

[1] https://sites.google.com/view/efficient-exploration

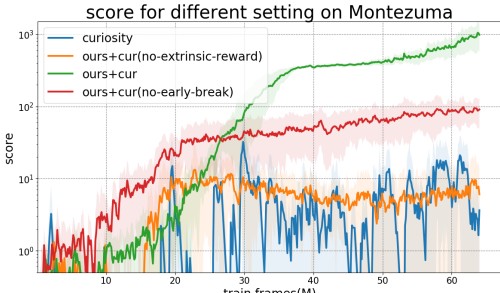 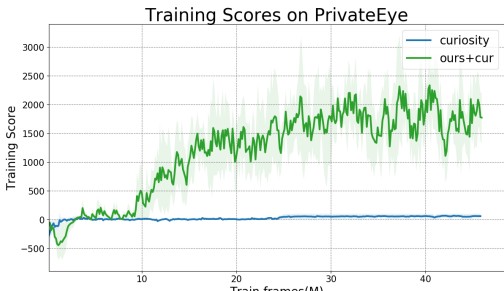

Figure 5:

As shown in Figure 5, in both `Montezuma's Revenge` and `PrivateEye`, our method outperforms the *curiosity* baseline with a noticeable gap.

In `Montezuma's Revenge`, we further implement multiple variants of Explicit Recall. The *no-extrinsic-reward* version can consistently get non—zero score (gets key in the game) in spite of no extrinsic reward is provided. The *no-early-break* version has more chance reaching selected target states, but also waste the lives when failing to follow the trajectory and deviates too much.

In `Montezuma's Revenge`, Comparing with pseudo-count (Bellemare et al., 2016) (their best result is obtained using DQN (Mnih et al., 2015), average score is 2461 at 50 million frames), the closest setting with us is A3C+ using deterministic environment (always accept action, fixed frame-skip) without life loss, their average score is 273.70 after 200 million frames. However, the result cannot be compared directly because of the different mechanism (count-based and curiosity) applied. But in `PrivateEye`, we achieve an average reward beyond 2000 within 50 million frames while their average reward of A3C+ is around 100 with 200 million frames.

# 4  RELATED WORKS

## 4.1  INTRINSICALLY MOTIVATED REINFORCEMENT LEARNING

Intrinsically motivated reinforcement learning for exploration is a set of approaches providing intrinsic rewards bonus using notions such as "unfamiliar", "novelty". There are many forms of the intrinsic reward function, such as count-based (Thrun, 1992; Strehl & Littman, 2008; Bellemare et al., 2016; Tang et al., 2016; Ostrovski et al., 2017; Martin et al., 2017), or related to predictive error (Stadie et al., 2015; Pathak et al., 2017), uncertainty (Schmidhuber, 1991; 2010; Houthooft et al., 2016; Lipton et al., 2018), etc. These approaches combines the intrinsic reward with extrinsic reward and optimized using reinforcement learning.

The major difference between our method and their approaches is the way of exploring. Where they encourage the agent to reach novel states by rewarding the agent, which is indirect. However, our approach is to let the agent directly guide itself towards novel states. We have shown the effectiveness of our method in experiments.

Our method is orthogonal to their methods as any of them can be used in the exploration phase of our method as a sub-procedure.

## 4.2  GOAL-DIRECTED EXPLORATION

Recently, another form of exploration has been proposed frequently. These methods generate goals through different ways and achieve the generated goals using a universal policy which generalizes over goal states (similar to UVFA Schaul et al. (2015a), HER Andrychowicz et al. (2017)). We categorize them as goal-directed exploration methods. Our method also falls into this category.

Held et al. (2017) proposes a framework that automatically generates proper goals using GAN Goodfellow et al. (2014) for the agent to achieve. Exploration is automatically driven by the generated goals and forms curriculum learning. Sukhbaatar et al. (2017) uses a framework named self-play,

which involves two agents Alice and Bob against each other. Bob wants to reproduce the task proposed by Alice. They use Alice's actions to generate goals. Baranes & Oudeyer (2013) proposes a framework SAGG-RIAC which also contains three stages: goal generation, achieving goals and further exploration. They generate goals by handcraft features.

The major difference between our method and their methods is the way of generating goals. Where they use generative neural networks, a sequence of actions, or handcraft features to propose goals to be reached. However, our method selects goals directly from former experience. There are several advantages to doing so. First, the selection is explicit, and our selected goals can easily be interpreted, as Figure 4 shows. Second, we do not need to "generate" goals which could be hard especially when the state space is high-dimensional, we just need to "select" the property goals from a set of options. Finally, and most importantly, explicit recallcan provide the agent itself its own experience (trajectory) of reaching those goals using memory. By having this trajectory, the problem of reach goals (might be hard in their method, especially when the horizon is long) becomes the easier one: following a trajectory. We split the goals into subgoals (similar to hierarchical reinforcement learning) using the trajectory, and avoid the problem of learning a long-term policy for reaching a state. Moreover, the trajectory provides the agent self-supervised signal on whether it successfully reproduce the state or not, encouraging the agent mastering interacting with the environment.

### 4.3 USE OF MEMORY

There are many approaches to exploit the power of memory to boost the performance of deep reinforcement learning algorithm, such as prioritized experience replay (Schaul et al., 2015b), neural episodic control(Pritzel et al., 2017).

Our method use memory to store the experience as trajectories into a prioritized pool. Although the usage of memory might be similar to prioritized experience replay (Schaul et al., 2015b) at first glance, but the usage of recalled trajectories is different to the replayed transitions in prioritized experience replay.

### 4.4 HIERARCHICAL REINFORCEMENT LEARNING

Hierarchical reinforcement learning (HRL) (Kulkarni et al., 2016; Vezhnevets et al., 2017) is a set of approaches that consisting of a hierarchical way of executing actions, one of the advantages of such approaches is that they are easier to deal with tasks of a long horizon.

Our way of following a trajectory by splitting it as subgoals can be viewed as a form of hierarchical reinforcement learning, where the PATH function is acted as a sub-policy.

## 5 CONCLUSIONS AND DISCUSSION

In this work, we proposed to utilize memory to improve efficiency of exploration in reinforcement learning. We developed Explicit Recall, a general mechanism that can be naturally combined with, and augment, many state-of-the-art exploration algorithms. It is shown to be effective in a number of simulated environments.

This work was inspired by cognitive studies Graziano (2006), which suggest humans also store the knowledge of movements or actions by their end-states, a process similar to our PATH function. Thus the PATH function can be viewed as a form of skills, and we provide a framework which could learn skills while doing pure exploration. There are also other ways to train the universal policy (skills) (Schaul et al., 2015a; Andrychowicz et al., 2017; Pathak et al., 2018). However, we believe that achieving self-generated tasks during spontaneous exploration and getting reinforced by self-supervised signals is a promising way for the agent to develop skills itself, and consider it as an interesting direction for further research.

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

# Supplementary Material for Explicit Recall

This supplementary material is organized as follows. First we introduce the setting of `Rooms` environment in Appendix A. Second, we explain in detail the *stage average reward* in Appendix B, which is used when extrinsic rewards of the trajectory are being considered. Then, we include the pseudo-code of the state reaching algorithm in Appendix C. Finally, we provided other implementation details and hyper-parameters in Appendix E.

## A   ROOMS ENVIRONMENT

### A.1   DEFINITION

Assuming the size of the map is $h \times w$, and the number of rooms is $n_h \times n_w$. First we use $\left\lfloor \frac{(h-1) \times i}{n_h} \right\rfloor$ to calculate the position of each separator rows (assume row $0$ and $h-1$ is the border). we can get columns similarly. There is one door connecting each pair of adjacent rooms in normal shape setting. Then we can calculate the position of doors between the rooms of both horizontal and vertical direction by averaging consecutive separators (rows or columns).

For the zigzag shape of the rooms, we keep all horizontal doors open but block vertical doors except for the rightmost doors between rooms of odd and even lines, and the leftmost doors between rooms from even and odd lines. Therefore, starting from the upper-left corner, to go down to the next line of rooms and explore the whole state space, the agent needs to move leftwards and rightwards interchangeably. The maximum distance between any two places in the environment is around $w \times (h_n - 1) + h$. Please check the upper-right part of the Figure 4 for the zigzag shape of $3 \times 3$ rooms of size $37 \times 37$.

### A.2   ENVIRONMENT SETTING FOR EXPERIMENTS

An episode ends when the agents reach the destination or have used 200 steps (except in zigzag shape $6 \times 6$ rooms, maximum steps is set to 300 to make sure the agent has the probability to explore the whole state space). When the destination is reached, the agent gets a reward of value 100. Otherwise, the agent gets no reward. When the agent hits the border or the walls, the action becomes invalid and has no effect.

The borders, walls and empty cells are represented by a single RGB value in visual representation. For pure exploration tasks, there is also a cell with different color indicating the current position of the agent. For finding destination tasks, another cell with different color indicates the destination in the visual state. The world is fully-observable.

In both tasks, the starting point is (2, 2) in all environment variants. The final point is fixed at (34, 34) for the destination finding task.

## B   STAGE AVERAGE REWARD FUNCTION

Here we use a simplified formula for the averaged reward. In our implementation, given a trajectory $(state_i, action_i, reward_i)$, the trajectory is first split by non-zero (where $reward_i \neq 0$) into several parts. Then each part calculates the average reward (*i.e.* total reward divide by the number of steps). Finally, the average rewards of each part get summed up and result in the *stage-average-reward*.

Due to the sparsity of reward signals of the environments we are tackling, we encourage the agent to use fewer steps to get the next reward signal. Therefore, we split the trajectory into stages by the reward signals and summed up the average reward in each stage. This also ensures that the exploring steps used on the current stage will not influence the score.

## C   ALGORITHM FOR STATE REACHING

---

**Algorithm 1** Algorithm for state reaching

---

**Inputs:**
A trajectory of states $traj$, the goal state is the last state of $traj$.
$env$: the environment that the agent interacts with.
$N$: maximum steps to the subgoal state on the $traj$.
$sim(x, y)$: similarity metric function for two states $x, \ y$.
$thresh$: a threshold constant.
$limit\_steps$: a function that determines maximum steps can take to achieve subgoals.
$g$: a function that converts similarity into inner reward.
**Output:** $(state, subgoal, action, inner\_reward)$
tuples which used for training the PATH function.
$state := env.reset()$
$i := 0.$
**while** $i < len(traj)$ **do**
  $n :=$ random integer in $[1, \min(N, len(traj) - i)]$.
  $i \mathrel{+}= n.$
  $subgoal := traj[i].$
  $step := 0.$
  $s := sim(state, subgoal).$
  **while** $s < thresh$ and $step < limit\_steps(n)$ **do**
    $action \sim$ PATH$(state, subgoal)$.
    $new\_state := env.act(action)$.
    $step \mathrel{+}= 1.$
    $s := sim(new\_state, subgoal).$
    $inner\_reward := g(s, thresh).$
    collect tuple $(state, subgoal, action, inner\_reward)$.
    $state := new\_state.$
  **end while**
**end while**

---

In the *no-early-break* setting, we let the agent try to recover the trajectory until the end of the trajectory. In all experiments, except mentioned as a variant, the limited number of consecutively missing subgoals is 20. The agent switches to the exploration phase if excesses the threshold.

`Limit_steps` imposes a limit on the number of steps when reaching a subgoal on a trajectory. Let $n$ be the number of steps used in the original trajectory. We take it as $3n/2$ for all experiments.

For the maximal length of the subgoal selection in Algorithm 1, we choose $N = 5$ for `Rooms` environments and $N = 15$ for Atari games. The inner reward function $g$ we used is $\left(\frac{c+1}{2}\right)^3$, where $c$ is the cosine similarity. The small penalty for training the PATH function is $-0.01$.

## D  HARDER EXPLORATION

We further exam the limit of our method in a more challenging variant of the `Rooms` environment. The environment contains $6 \times 6 = 36$ rooms, and the only way to the lower-right corner from the upper-left corner is by following the zigzag shape route. We include the demo video on our website. Shown in Figure 6, the *ours+count* method is still able to fully explore the state space.

## E  IMPLEMENTATION DETAILS AND HYPER-PARAMETERS

During the most recent exploration trials, the experience starting from the beginning of the episode until current state is periodically adding to the memory every 10 steps. Meanwhile, in the case of some states are hard to reach, every state added to the memory have 10 copies, but the capacity of the memory *pool* only count the unique ones.

For Atari games, We use OpenAI gym wrapper (Brockman et al., 2016) as our environment. When learning the inverse dynamics model $\phi$, the agent should know the action actually applied, thus we

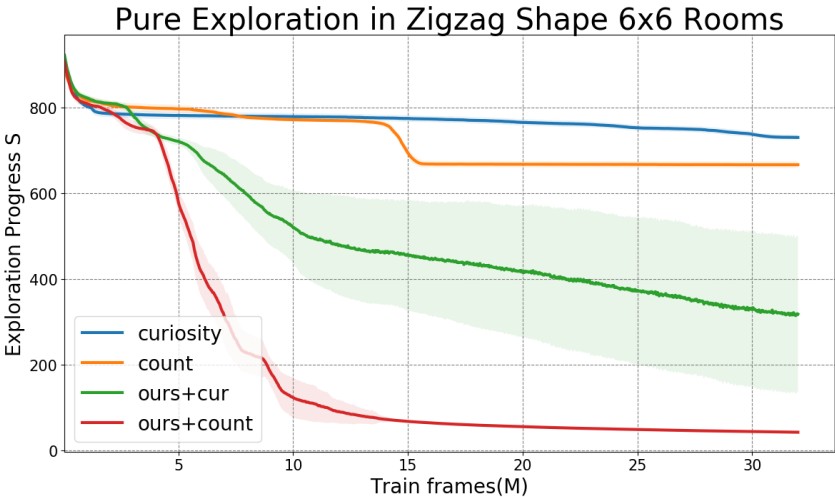

Figure 6: 6x6 Zig Shape.

let the actual actions be consistent with the chosen actions in the experiments (*i.e.* use v4 instead of v0 for gym environments).

In Montezuma's Revenge, to avoid the redundancy in action space influencing the training of inverse dynamics model $\phi$, we shrink the available action set (*e.g.* remove *UP-LEFT*). This modification doesn't have a significant influence on the overall performance except for the small degradation in *ours+cur*, where raw A3C as the explorer benefits a bit from the shrinkage of the action space.

We use a variant implementation of A3C called batched-A3C, and preserve its parameters. We use 50 players for A3C, each with a separate experience pool, the size is 200 (unique trajectories) across all experiments. We use brute-force to maintain the experience pool (as this part is not the speed bottleneck of the whole method). To avoid calculating the unfamiliarity too frequently, we decay the calculated unfamiliarity (by a factor of 0.996) after each episode instead of recalculating them.

For neural network structures, we use the same one as (Pathak et al., 2017) except that the number of layers of $\phi$ is 3 instead of 4.

