# OpenReview forum: "Explicit Recall for Efficient Exploration"
_ICLR.cc/2019/Conference_

### Official Review · AnonReviewer2 · 2018-10-15
**Many hacks and heuristics.**

**Rating:** 3
**Confidence:** 4

**Review:**

In this paper, the authors propose a heuristic method to overcome the exploration in RL. They store trajectories which result in novel states.
The final state of the trajectory is called goal state, and the authors train a path function which given a state and a subgoal states (some states in the trajectory) the most probably action the agent needs to take to reach the subgoal. These way they navigate to the goal state. The goal state is claimed to be achieved if the feature representation stoping state is close to goal (or subgoal for subgoal navigation).


The authors mainly combine a few previous approaches "Self-Imitation Learning," "Automatic Goal Generation for Reinforcement Learning Agents," and "Curiosity-driven exploration by self-supervised prediction" to design this algorithm which makes this approach less novel.

General comment; there are variable and functions in the paper that are not defined, at least at the time, they have been used. The Rooms environment is not described. What is visit_times[x] and x is not a wall? What is stage avg reward? and many others

The main idea of the algorithm is clear, but the description of the pieces is missing.

It is not clear in stochastic setting how well this approach will perform.

The authors state that
"Among different choices of the modeling, we choose inverse dynamics (Pathak et al., 2017) as the environment model, which has been proved to be an effective way of representing states under noisy environments."
I took a look at this paper and could not find neither proof or quantification of "effective"-ness. Please clarify what the meaning this statement is.

Why s=s' is ambiguous to the inverse dynamics?

What is the definition of acc in fig2?

why (consin+1)^3/8 is chosen?

---

> ### Author Response · Authors · 2018-12-04
> **Our Response**
>
> Thanks for your comments and suggestions.
>
> 1. Comparison with previous methods.
> Different from “Self-Imitation Learning”, our agent uses explicit memory to help exploration, whose advantages are described in the last paragraph of page 1.
> The major difference between our method and "Automatic Goal Generation for Reinforcement Learning Agents" is the way to generate the goals and is stated in details in Section 4.2.
> The notion of curiosity defined in “Curiosity-driven exploration by self-supervised prediction” is employed in our framework for the exploration. Comparison has been made with the ICM model proposed in this paper. See Figure 1-3 and 5.
>
> 2. The definitions.
> The details about Rooms environment can be found in Appendix A. The visit_times[x] means the number of times the cell x is being visited by the agent, accumulated throughout the training process. In the Rooms environment, each cell has a type of empty/wall/border, The stage avg reward is used as a metric for evaluating the trajectories, whose details can be found in Appendix B. We will try to integrate these definitions into the main text.
>
> 3. Performance in stochastic setting
> Both Montezuma’s Revenge and PrivateEye environments are stochastic: each action leads to 2~4 frame-skips randomly. Our method outperforms the curiosity baseline in both environments. As for random starting states, please see the response to AnonReviewer3 (A. About the same-start assumption)
>
> 4. Clarifications
> A. We would change the words to make it more clear. Here what we mean is that the inverse dynamics provides a feature space that ignores the noise which the agent cannot control (e.g. white noise in visual input) (as suggested by Pathak et al., 2017).
> B. When there are multiple actions leading to the same next state s’, the inverse dynamics would have multiple answers. This is what we mean “ambiguous”.
> C. In Fig2, the accuracy is the number of cases that the output \hat{a} leads to the desired next state s’, that is, env(s, hat(a)) = s’.
> D. (cos+1)^3/8 is chosen empirically, used for modeling the similarity between states.

---

### Official Review · AnonReviewer3 · 2018-10-17
**Interesting idea, but rather weak paper. Can be improved a lot with additional writing effort**

**Rating:** 4
**Confidence:** 4

**Review:**

In this paper, the authors propose an exploration strategy based on the explicit storage and recall of trajectories leading to novel states. A pool of such trajectories is managed over time, and a method is proposed so that the agent can learn how to follow a path corresponding to these trajectories so as to explore novel states. The idea is demonstrated in a set of room experiments, and quickly shown efficient in Montezuma's Revenge and PrivateEye Atari games.

Overall, the idea has some merits, but the empirical study is weak and the paper suffers from unsufficient writing effort (or more probably time).

What I like most in the paper is the split of exploration methods into 3 categories: adding some "intrinsic reward" bonuses to novel states (curiosity-driven exploration) , trying to reach various goals (goal-driven exploration) and using memory to reach again novel states (memory-driven exploration). Actually, this split may be debated. For instance, some frameworks based on goals have been labelled curiosity-driven, e.g. "Curiosity-Driven Exploration of Learned Disentangled Goal Spaces" (Laversanne-Finot, Péré and Oudeyer, CoRL 2018), but anyways I find it useful. That said, this aspect of the introduction is reiterated in the "Related Work" section in a quite redundant way, whereas both parts could have been better integrated. Furthermore, the related work section is hardly a draft, I'll come to that later.

The presentation of the method in Section 2 is rather clear and convincing. My only concern is about the assumption that the agent is always starting in the same state. This assumption may not hold in many settings, and the approach appears to be quite dependent on it. A discussion of how it could be extended to a less restricting assumption would be welcome.

The experimental section is weaker. A few concerns:
- I could not find much about the number of seeds, trials, the way to depict some variance, the statistical significance of the differences between results presented in Figure 1. The same is true about Figs. 2, 3 and 5.
- In Fig.2, the claim that the author's method learns better models is hardly supported by the left-hand-side plot, and significance is not assessed.
- I'm puzzled about the very low performance of baselines in the plots of Fig. 3. Could the author explain why these performances are null.
- The Atari games section helps figuring out that the framework is not too specific of the rooms environment, but the lack of analysis does not help making sure that this is just the explicit recall mechanism that is responsible for superior performance and why.


Another point about this section is that poor writing does not help understanding some points.
- to me, the first sentence of Section 3.2.2 does not make sense at all.
- in the caption of Fig. 4, "The second row is the heatmaps for states that the number of times being selected as a target state.": I don't get what it means, thus I don't understand what that row shows.
- Fig.5 comes with no caption

About the related work:
- The comparison to other methods using memory needs to be expanded. In particular, I would put HER-like mechanisms here rather than in 4.1, as "explicit recall" shares some importan ideas with "experience replay"
- Section 4.4 (HRL) is not useful as is.

Finally, in the conclusion, the claim that the method can be combined with "many sota exploration methods" is not supported, as the authors have only tried two and did not analyse the results in much details.


typos:

- p4:
we can easily counting
(include borders) => including
is provide => provided

are less less-visited states: quite inelegant

- p7:
In Montezuma's Revenge, Comparing => comparing
Where they encourage => remove "Where"

- p8:
recallcan => recall can
the problem of reach goals => reaching
it succesfully reproduce => reproduces

The last paragraph of Section 4.2 needs a careful rewriting, as long sentences with parenthese in the middle appear to be some draft version.

control(Pritzel => Missing space
Our method use memory => uses
Although ..., but => remove but

The path function can be seen as a form of skills => skill?
Besides, the "can be seen" needs to be further explained...

Appendix

Finally, we provided => provide

is around (math formula) => cannot you be more specific?

---

> ### Author Response · Authors · 2018-12-04
> **Our Response**
>
> Thanks for your comments and suggestions, and we will revise the paper as you suggested.
>
> 1: About the same-start assumption.
> We discuss the starting states in four cases.
> A. The starting states are always the same, which is our the assumption.
> B. There is a small randomness (noise) for the starting state. Path function can handle this: after choosing a goal state from a trajectory, Path function will generate a trajectory from the current starting state to the goal state.
> C. There are multiple possible starting states. New episodes can start in the same states as *some* (not all) previous episodes: the agent can simply remember successful trajectories and apply our algorithm to distinct start states separately.
> D. If the starting states are too far away (or randomly given) and no assumption is made about their relation/similarity, little can be expected to take advantage of former trajectories, even for humans.
>
> 2: Experiment details
> A. The number of seeds is 2 for experiments in Rooms environment, 3 for Atari Games.
> B. Re Fig2: better exploration in RL is expected to lead to a faster learning curve, not necessarily a better final model.  Fig 2 shows exactly this: our method learns faster than the baseline, without sacrificing the final model performance.
> C. Re Fig3: as the destination are very far away from the starting point (see Appendix A.2), agents’ score would be almost 0 if the destination could not be found during the exploration. The Zigzag-shaped rooms environment requires the agents to explore the full map to reach the destination. The results are consistent with Fig1 showing the exploration efficiency.
>
> 3: Clarification on technical details.
> We apologize for any confusions in the paper and will improve the writing. Specific questions by the reviewer are addressed as the following:
> On the first sentence of Section 3.2.2. While Path function (we regard it as skills) is being trained independently with the task, it can be applied on any other tasks. For example, in Zero-shot Visual Imitation [1], the goal-conditioned policy is used to follow a sequence of key-points demonstration.
> In Fig 4, the second row shows the number of times each state being chosen as the target state (the last state of a selected trajectory). This number is illustrated as a heatmap with log-scale.
>
> [1] Deepak Pathak, Parsa Mahmoudieh, Guanghao Luo, Pulkit Agrawal, Dian Chen, Yide Shentu, Evan Shelhamer, Jitendra Malik, Alexei A. Efros, and Trevor Darrell. Zero-shot visual imitation. In ICLR, 2018.

---

### Official Review · AnonReviewer1 · 2018-11-05
**Good idea, good demonstration, good score**

**Rating:** 7
**Confidence:** 3

**Review:**

This paper is the first showing that achieving self-generated tasks during spontaneous exploration and getting reinforced by self-supervised signals is a promising way for the agent to develop skills itself.
The scores are demonstrative on several tasks.
It opens interesting direction for further research.

REM:
few typos like "An state"
Please plot in dash the count methods in the graphs (use oracle information)

Annexe C shall be integrated into the core of the paper. Could be simplified.
The cosine metrics shall be better integrated in it.

---

> ### Author Response · Authors · 2018-12-04
> **Our Response**
>
> Thanks for your encouraging comments and nice suggestions. We plan to update the figures in the paper upon the decision. We will also integrate Appendix C and the cosine metrics into the main text.

---

### Public Comment · ~Anirudh_Goyal1 · 2018-11-27
**More references**

Hello,

I just came across your paper. I think few other papers should be cited, which also tries to use explicit memory in terms of high value states or goal states or high bellman error.

[1] Recall Traces, https://arxiv.org/abs/1804.00379  (I'm the author of this paper)
[2] Self Immitation learning, https://arxiv.org/abs/1806.05635
[3] Neural episodic control https://arxiv.org/abs/1703.01988

Thanks for your time! :)

---

> ### Author Response · Authors · 2018-12-04
> **Thanks**
>
> Thanks for your pointers to the related works. We will definitely add them to our references and compare with them upon the decision.

---

### Author Response · Authors · 2018-12-04
**Comparison with Go-Explore**

Recently, a similar method is published on uber’s website (https://eng.uber.com/go-explore/), which they called the go-explore method. Their results are very promising on both Montezuma’s Revenge and Pitfall, two of the hardest exploration tasks in Atari games.

While we share the similar 3-stage exploration structure, there are several differences.
1. As they assume the environment is resettable/deterministic during training, they can utilize the ability of reset to quickly return to a state the agent want. Instead, we do not rely on the assumption, which brings significant hardness while reaching an intended state, and is the major reason why our performance is not as good as theirs.
2. When the training environment is stochastic (in our montezuma’s setting), they propose to use goal-conditioned policy, which is exactly what we are doing. Furthermore, we also propose to sample sub-goals from the trajectory.

---

### Meta-Review · Area_Chair1 · 2018-12-13
**Interesting research direction but weak paper**

**Confidence:** 4
**Recommendation:** Reject

**Metareview:**

The paper presents an explicit memory that directly contributes to more efficient exploration. It stores trajectories to novel states, that serve as training data to learn to reach those states again (through iterative sub-goals).

The description of the method is quite clear, the method is not completely novel but has some merit. Most weaknesses of the paper come from the experimental section: too specific environments/solutions, lack of points of comparisons, lacking some details.

We strongly encourage the authors to add additional experimental evidence, and details. In its current form, the paper is not sufficient for publication at ICLR 2019.

Reviewers wanted to note that the blog post from Uber ("Go-Explore") did _not_ affect their evaluation of this paper.